



# Global riverine flood risk – how do hydrogeomorphic floodplain maps compare to flood hazard maps?

Sara Lindersson[1,2], Luigia Brandimarte[3], Johanna Mård[1,2], Giuliano Di Baldassarre[1,2]

[1]Centre of Natural Hazards and Disaster Science (CNDS), c/o Department of Earth Sciences, Uppsala University, 75236, Uppsala, Sweden
[2]Department of Earth Sciences, Uppsala University, 75236, Uppsala, Sweden
[3]KTH – Royal Institute of Technology, 10044 Stockholm, Sweden

*Correspondence to*: Sara Lindersson (sara.lindersson@geo.uu.se)

**Abstract**

Riverine flood risk studies require the identification of areas prone to potential flooding. This process can be based on either (hydrologically-derived) flood hazard maps or (topography-based) hydrogeomorphic floodplain maps. In this paper, we derive and compare riverine flood exposure from three global products: a hydrogeomorphic floodplain map (GFPLAIN) and two

flood hazard maps (JRC and GAR). We find an average spatial agreement between these maps of around 30 % at river basin level on a global scale. This agreement is highly variable across model combinations and geographic conditions, influenced by climatic humidity, river volume, topography, and coastal proximity. Contrary to expectations, the agreement between the two flood hazard maps is lower compared to their agreement with the hydrogeomorphic floodplain map. We also map riverine flood exposure for 26 countries across the Global South, by intersecting these maps with three human population maps (GHS,

HRSL and WorldPop). The findings of this study indicate that hydrogeomorphic floodplain maps can be a valuable way of producing high-resolution maps of flood-prone zones to support riverine flood risk studies, but caution should be taken in regions that are dry, steep, very flat or near the coast.

## 1 Introduction

Flood disasters are a major cause of loss throughout the globe, claiming thousands of lives and causing substantial economic

damage every year (CRED and UNDRR, 2020). Our ability to map population growth within flood-prone zones is important since increased exposure is a key driver of flood risk (Ceola et al., 2014; CRED and UNDRR, 2020; Jongman et al., 2012; Winsemius et al., 2016). Global maps of flood-prone zones and human settlements are useful for detecting risk hotspots across the world, and may also be used for local studies in data-poor regions (UN SDSN, 2020; Ward et al., 2020). Open-access global maps are providing a variety of alternative products (Lindersson et al., 2020), navigating among these offers might be

challenging to the users.


Riverine flood risk studies traditionally outline flood-prone zones with hazard maps from hydrodynamic or hydraulic models (Ward et al., 2020). These hazard maps typically show flood extent corresponding to a certain probability, for example, the 100-year flood (meaning it has a 1 % chance to occur any given year). This is, however, a computationally demanding method that requires a lot of data: estimating river flow corresponding to a certain probability requires long time series of

meteorological or hydrological data, which is a rarity (Blöschl et al., 2013; Kidd et al., 2017). To subsequently map the two-dimensional water extent with an inundation model requires a detailed topographic model, and information about the river profile and surface roughness (Dottori et al., 2013; Hunter et al., 2007).

Hydrogeomorphic methods for mapping floodplains, on the other hand, distinguish the characteristic shapes of floodplains based on topography (Dodov and Foufoula-Georgiou, 2006; Nardi et al., 2006). Moreover, hydrogeomorphic terrain analysis

is computationally efficient and scale-invariant, the floodplain maps can without difficulty be renewed whenever refined terrain models become available (Manfreda et al., 2014; Nardi et al., 2018; Tavares da Costa et al., 2019). Yet, a floodplain map does not provide the user with any information about flood extent probability, which is often needed in flood risk applications (Dottori et al., 2016; Sampson et al., 2015). Annis et al. (2019) and Tavares da Costa et al. (2019) suggest that hydrogeomorphic models can be useful for large-scale floodplain mapping in ungauged basins, in locations where reliable flood hazard maps are

unavailable. Di Baldassarre et al. (2020) argue that flood hazard maps and hydrogeomorphic floodplain maps are complementary and should both be used for identifying flood-prone zones in data-scarce regions, following the precautionary principle (Foster et al., 2000). The recently developed global floodplain map GFPLAIN250m (Nardi et al., 2019), hereafter GFPLAIN, built by hydrogeomorphic terrain analysis, has raised interest due to its potential for large-scale riverine flood risk applications (Akhter et al., 2021; Mazzoleni et al., 2020; Nardi et al., 2019).

What we know, however, is that the results of large-scale flood exposure analysis heavily depend upon the datasets used (Aerts et al., 2020; Dottori et al., 2016; Smith et al., 2019; Trigg et al., 2016; Ward et al., 2020). This is exemplified in the work undertaken by Trigg et al. (2016) showing that outputs from six individual global flood hazard models yield considerably different exposure estimates for the African continent. While finding many areas of agreement, particularly in connection to large rivers with distinct floodplain boundaries, the overall model agreement was only 30 to 40 % for the entire African

continent (Trigg et al., 2016). The flood models particularly disagreed in arid regions, deltas, and large wetlands (Trigg et al., 2016). The floodplain map GFPLAIN has not been included in large-scale comparison studies, however, so the suitability of GFPLAIN compared to other global flood hazard maps for analysing riverine flood exposure remains unclear.

Previous work comparing hydrogeomorphic floodplain maps with flood hazard maps have primarily been limited to local and regional case studies. Nardi et al. (2019) and Tavares da Costa et al. (2019) quantitatively compared the consistency between

GFPLAIN and global flood hazard maps for several European rivers. Annis et al. (2019) found that the consistency between a hydrogeomorphic floodplain map and a flood hazard map was affected by the interplay of terrain model resolution, terrain analysis scaling parameter, and river stream order. Furthermore, the consistency increased with increasing return period of the flood hazard map, meaning that the hydrogeomorphic model delineates floodplains generated by high-magnitude, low-





frequency events (Annis et al., 2019). Nardi et al. (2018) analysed how well hydrogeomorphic floodplain results mimicked

flood hazard maps, and predominantly found consistencies in floodplain areas unaltered by humans.

In this study, we build upon the existing literature comparing outputs of global flood models, by examining how the global floodplain map GFPLAIN (Nardi et al., 2019) compares to two commonly used global flood hazard maps when estimating riverine flood exposure (Dottori et al., 2016; CIMA Foundation, 2015). The aim of our analysis is to (a) examine the spatial agreement between GFPLAIN and global flood hazard maps across a range of geographic conditions, and (b) demonstrate

how model differences affect the estimation of riverine flood exposure. We performed this comparison in three steps. First, we quantified the model agreement for all the river basins of the world covered by all three models. Second, we controlled if the model agreement is associated with specific hydro-environmental attributes. Third, we mapped riverine flood exposure for 26 countries across the Global South, by intersecting the maps of flood-prone zones with three individual population datasets. The purpose of this paper is to shed light on the usability of hydrogeomorphic floodplain maps in flood risk studies, and how

the usability varies across geographic conditions.

## 2 Data

This section describes the datasets used for comparing the usability of a global hydrogeomorphic floodplain map to two flood hazard maps for riverine flood exposure analysis. A technical summary of the flood and population maps can also be found in Table A1.

### 2.1 Flood maps

We represent riverine flood hazard by comparing three models of flood-prone zones: the global floodplain map GFPLAIN (Nardi et al., 2019), the Flood Hazard Map of the World by the European Commission's Joint Research Centre, hereafter JRC (Dottori et al., 2016), and the flood hazard maps produced for the Global Assessment Report on Disaster Risk Reduction 2015, hereafter GAR (CIMA Foundation, 2015). The purpose of this comparison is to analyse how GFPLAIN, derived with

hydrogeomorphic terrain analysis, represent riverine flood-prone zones compared to the flood hazard models of JRC and GAR. There are currently several outputs from global flood models available, as exemplified by the hazard maps of CaMa-Flood (Yamazaki et al., 2011), GAR (CIMA Foundation, 2015), Fathom Global Ltd (Sampson et al., 2015), JRC (Dottori et al., 2016), and GLOFRIS (Winsemius et al., 2016). We used JRC and GAR since they are open-access datasets and are, like GFPLAIN, based on the global elevation model Shuttle Radar Topography Mission (SRTM) (Farr et al., 2007; Reuter et al.,

2007). This allows for a comparison of methods delineating flood-prone zones, rather than a comparison of underlying terrain model performance. Furthermore, JRC and GAR have been in high demand in flood exposure studies (Alfieri et al., 2017; Ehrlich et al., 2018; UNISDR, 2015; Zischg and Bermúdez, 2020), and included in previous inter-model comparison and validation studies (Aerts et al., 2020; Bernhofen et al., 2018; Trigg et al., 2016). The GAR model has also been referred to by previous literature as the CIMA-UNEP model (Bernhofen et al., 2018; Trigg et al., 2016).



The model structure of JRC and GAR differs in the sense that GAR builds upon one-dimensional hydraulic modelling, while JRC builds upon two-dimensional hydrodynamic modelling (Dottori et al., 2016; CIMA Foundation, 2015). The model chain of JRC used 33 years of reanalysed climate data (ERA-Interim) to calculate the probability of each pixel being flooded (Dottori et al., 2016). The model of GAR used statistical regionalization of gauged streamflow observations to calculate extreme flow values, which were then used as input to a hydraulic model for flood inundation mapping (CIMA Foundation, 2015). The maps

of JRC differ from GFPLAIN and GAR in river network coverage: GFPLAIN and GAR include rivers with upstream drainage areas larger than 1000 km$^2$, while the corresponding threshold for JRC is 5000 km$^2$ due to the coarse spatial resolution of the climate data (Dottori et al., 2016; Nardi et al., 2019; CIMA Foundation, 2015).

The flood model of JRC does not take into account flood protection infrastructure, overbank flow tends to result in floodplain inundation due to unconstrained lateral extents (Dottori et al., 2016). This should favour agreement with the floodplain

delineation of GFPLAIN since the hydrogeomorphic terrain method does not capture disrupted connectivity between the river channel and floodplain either. The GAR hazard maps, on the other hand, have been post-processed based on the assumption that the design level of flood defence is a function of the maximum GDP of the area (CIMA Foundation, 2015). The modelled defence levels in GAR were assumed to partially fail if the design return period was exceeded (CIMA Foundation, 2015).

The morphology of rivers, which the hydrogeomorphic models aim to capture, is predominantly shaped by high-magnitude,

low-recurrence flood events (Annis et al., 2019; Bhowmik, 1984; Nardi et al., 2018). We, therefore, conducted all analyses in this study using the JRC and GAR hazard maps with return periods 100- and 500-years (hereafter JRC-100, JRC-500, GAR-100 and GAR-500).

The spatial resolution of GFPLAIN is 8.33 arcseconds (~250 m near the line of the equator) and the spatial resolution of JRC and GAR hazard maps are 30 arcseconds (~1 km). The GFPLAIN dataset is provided as one raster file per continent, covering

a near-global extent (60° N, 56° S). JRC is offered as global (85° N, 85° S) seamless raster files; raster files of GAR also cover the near-global extent (60° N, 56° S).

## 2.2 Population maps

Throughout this paper, we use the term flood exposure to refer to the number of people located within flood-prone zones. Exposure analysis was conducted by intersecting GFPLAIN and JRC with three individual population maps: the High

Resolution Settlement Layer, hereafter HRSL (Facebook Connectivity Lab and CIESIN, 2016), Global Human Settlement Population Layer, hereafter GHS (Schiavina et al., 2019) and the WorldPop population layer, hereafter WorldPop (Gaughan et al., 2013; Linard et al., 2012; Sorichetta et al., 2015; Tatem, 2017).

We chose to conduct the exposure analysis with these population datasets since they represent diverse methodologies for mapping populations, each having individual advantages and limitations. We briefly describe these differences here, other

studies have reviewed how the choice of population dataset affects exposure analysis (Leyk et al., 2019; Smith et al., 2019). First of all, the individual population maps offer a range of spatial resolutions: HRSL is 1 arcsecond (~30 m), WorldPop is 3 arcseconds (~90 m) and GHS is 9 arcseconds (~250 m).





Dasymetric mapping refers to, in this case, the spatial reallocation of census data to individual pixels using information from ancillary data (Leyk et al., 2019; Wright, 1936). GHS used a binary dasymetric approach to map population count, allocating

census data to built-up areas detected with Landsat satellite imagery (Schiavina et al., 2019). HRSL used convolutional neural networks for allocating census data to buildings that have been detected with DigitalGlobe high-resolution satellite imagery (Smith et al., 2019). WorldPop used a multivariate dasymetric approach to allocate population data to settlements detected with Landsat satellite imagery, with multiple ancillary data layers, e.g. land cover, built structures, infrastructure, travel time to major cities, nighttime lights (Lloyd et al., 2017). The multivariate modelling approach of WorldPop means that these

covariate variables might prevent interaction studies.

The main asset of GHS is the temporal depth of the dataset, enabling change detection analysis with its globally consistent population grids for 1975, 1990, 2000 and 2015 (Ehrlich et al., 2018). WorldPop currently offers population count maps for every year between 2000-2020. The rather coarse spatial resolution of Landsat means, however, that GHS and WorldPop tend to leave out dispersed rural settlements (Smith et al., 2019; UN SDSN, 2020). HRSL has shown to better represent settlements

in rural areas (Smith et al., 2019; Tiecke et al., 2017) but is, however, presently only available as a static dataset.

The population maps of HRSL and WorldPop are provided as one raster file per country, HRSL covers ~140 countries while WorldPop is global. GHS is offered as a global (85° N/85° S) seamless raster file. Population maps used in this study represent the year 2015 (GHS) and 2018 (HRSL and WorldPop).

## 3 Methods

### 3.1 Data homogenization

The geospatial analysis has primarily been conducted using the cloud computation platform Google Earth Engine (Gorelick et al., 2017). GFPLAIN was tailored for analysis by merging the individual continental images into one single image. GFPLAIN, JRC and GAR were then reclassified to binary wet/dry maps. Normally wet areas were masked from GFPLAIN, JRC and GAR with the global water mask MOD44W, a product that combines waterbody data from SRTM and the satellite imagery of

MODIS (Carroll et al., 2009). MOD44W shares the spatial resolution with GFPLAIN, 8.33 arcseconds (~250 m).

The individual country images of HRSL and WorldPop were also merged into seamless global images, using the median pixel value where multiple images overlap. The metadata of the population maps were filtered to the year 2015 for GHS and the year 2018 for HRSL and WorldPop.

### 3.2 Model agreement

To quantify the model agreement between GFPLAIN, JRC and GAR, we used the Model Agreement Index (MAI), Eq. (1), as proposed by Trigg et al. (2016). This agreement index allows for an evaluation of multiple raster files. The maps from all three models were first aggregated (separately for each return period) into categories based on how many models agree that each pixel is flooded: 0 = all models are dry, 1 = one model is wet, 2 = two models are wet, 3 = all models are wet. This aggregation



was conducted at the finest spatial resolution of 8.33 arcseconds. MAI values, Eq. (1), were then calculated for all the basins
in the world that are covered by all three models, resulting in 2776 river basins. By only including the basins covered by all
three flood models we ensured that differences in spatial coverage did not affect the results. For instance, the individual flood
maps have been post-processed to mask arid areas, to different degrees, since aridity poses a challenge for traditional flood
model assumptions. The HydroBasins Level 5 dataset, based on 15 arcseconds resolution raster data, was used for outlining
the river basin boundaries (Lehner and Grill, 2013). The output value of MAI, Eq. (1), ranges between 0 (no agreement) and
1 (full agreement):

$$MAI = \frac{\sum_{i=2}^{N} \frac{i}{N} a_i}{A}, \tag{1}$$

where; $A$ is the total number of flooded pixels by all three models, $a_i$ is the number of pixels flooded by that particular
aggregated category, $N$ is the number of models in comparison, and $i$ is the number of models in agreement for that particular
category. This index does not assume that one model is preferable to the other and also ignores the large areas that are marked
as dry, which would otherwise bias the performance measure. MAI values, Eq. (1), evaluating the agreement of the models
GFPLAIN, JRC and GAR were calculated for all 2776 river basins, for the return periods 100- and 500-years separately,
hereafter MAI-500 and MAI-100.

Following this, we also quantified pairwise model agreement for each combination of flood model, Eq. (2). The pairwise model
agreement in Eq. (2) corresponds to Eq. (1) when letting $N = 2$, and has frequently been recommended for evaluating binary
maps of inundation models (Aronica et al., 2002; Bates and De Roo, 2000; Pappenberger et al., 2007; Schumann et al., 2009;
Trigg et al., 2016). The agreement index in Eq. (2), also referred to as an $F^2$ measure (Aronica et al., 2002; Pappenberger et
al., 2007) or F-index (Annis et al., 2019), ranges between 0 (no agreement) and 1 (full agreement):

$$MAI_{N=2} = \frac{a}{a + b + c}, \tag{2}$$

where; $a$, $b$ and $c$ denote the number of pixels within each river basin according to the contingency table in Table 1. Each
combination of GFPLAIN, JRC and GAR was evaluated with Eq. (2) for all 2776 river basins, for the return periods 100- and
500-years separately: MAI_GFPLAIN GAR-100, MAI_GFPLAIN GAR-500, MAI_GFPLAIN JRC-100, MAI_GFPLAIN JRC-500, MAI_GAR-100 JRC-100 and
MAI_GAR-500 JRC-500.

**Table 1: Contingency table for the pairwise model agreement evaluation as given by Eq. (2). The variables *a*, *b*, *c* and *d* relate to the
number of pixels within each river basin.**

|  | Within Map 1 | Outside Map 1 |
|---|---|---|
| Within Map 2 | *a* | *b* |
| Outside Map 2 | *c* | *d* |



### 3.2.1 Spatial agreement clusters

We identified spatial clusters of basins with high and low model agreement relative to the mean, i.e. hot- and cold spots, using the MAI-500 value for each river basin. This local spatial autocorrelation analysis was carried out in the software GeoDa, version 1.18 (Anselin et al., 2006). Positive spatial autocorrelation was found in high-agreement basins also having high-agreement neighbouring basins, or low-agreement basins with low-agreement neighbours. More specifically, we used a local Moran statistic (Anselin, 1995), with 9999 random permutations, to identify significant clusters with pseudo p-values > 0.05.

The significant clusters were mapped together with the original MAI-500 values to exhibit global patterns of the model agreement between the three flood models.

### 3.2.2 Hydro-environmental attributes

The association between MAI-500 and several hydro-environmental attributes was then quantified for all 2776 river basins using statistic techniques in R (R Core Team, 2014). Basin level data for all attributes were retrieved from the datasets

BasinATLAS version 1.0 (Linke et al., 2019) and HydroBasins Level 5 (Lehner and Grill, 2013). Non-parametric measures were used for this evaluation since the response variable, MAI-500, did not fulfil the normality assumption of general linear models. An alpha level of 0.05 was used for all statistical tests.

The Kruskal-Wallis test by ranks (Hollander and Wolfe, 1973) was performed to control differences in MAI-500, Eq. 1, for the following factors: geographic region, river stream order (Lehner and Grill, 2013), and freshwater major habitat type (Abell

et al., 2008). The freshwater major habitat types entail a combination of topographic, climatic and hydrological properties since this classification is based on similarities in biological, chemical and physical characteristics (WWF and TNC, 2019). All Kruskal-Wallis tests were supplemented with post hoc Wilcoxon rank-sum tests, performing pairwise comparisons between the groups. The Kruskal-Wallis tests used the Benjamini and Hochberg (1995) correction method for controlling false discovery rate. We also performed Kruskal-Wallis and Wilcoxon rank-sum tests to evaluate differences between the pairwise

model agreement distributions ($MAI_{GFPLAIN\ GAR-100}$, $MAI_{GFPLAIN\ GAR-500}$, $MAI_{GFPLAIN\ JRC-100}$, $MAI_{GFPLAIN\ JRC-500}$, $MAI_{GAR-100\ JRC-100}$ and $MAI_{GAR-500\ JRC-500}$).

We then quantified the level of association between the model agreement and individual hydro-environmental attributes. Spearman rank-order correlation coefficients (Harell Jr and Dupont, 2021; Hollander and Wolfe, 1973; Spearman, 1904) were calculated for MAI-500, $MAI_{GFPLAIN\ GAR-500}$, $MAI_{GFPLAIN\ JRC-500}$, $MAI_{GAR-500\ JRC-500}$ and 23 control variables listed in Table 2.

The resulting correlation coefficients were plotted in a correlogram, ordered according to a hierarchical clustering method (Wei and Simko, 2017).




**Table 2 Control variables for which the association with the model agreement between the flood maps was evaluated. All control variables were retrieved from the dataset BasinATLAS version 1.0 (Linke et al., 2019), except river stream order and basin area, being imported from the HydroBasins dataset (Lehner and Grill, 2013). The upstream watershed is the total upstream watershed directly connected to the basin.**

| Attribute | Description | Unit | Spatial dimension | Source |
|---|---|---|---|---|
| River stream order | Classification according to a classical river ordering system. Order 0 is used for coastal basins. | Classes (4) | Basin classification | HydroBasins (Lehner and Grill, 2013) |
| Freshwater major habitat types | Classification based on freshwater ecoregions. | Classes (11) | Spatial majority | Freshwater Ecoregions of the World (Abell et al., 2008) |
| Basin area | Area of the individual basin, and the upstream watershed respectively. | Square kilometres | Sum | HydroBasins (Lehner and Grill, 2013) |
| Natural discharge annual | Long-term (1971-2000) naturalized discharge at the basin pour point. | Cubic meters per second | Annual average | WaterGAP v2.2 (Döll et al., 2003) |
| Land surface runoff annual | Long-term (1971-2000) naturalized runoff within basin. | Millimetres | Annual average | WaterGAP v2.2 (Döll et al., 2003) |
| River volume | River volume within the basin and the upstream watershed respectively. | Thousand cubic meters | Sum | HydroSHEDS & WaterGAP (Lehner and Grill, 2013) |
| Elevation average | Terrain elevation average within the basin outlines. | Meters above sea level | Average | EarthEnv-DEM90 (Robinson et al., 2014) |
| Slope average | Terrain slope average within the basin outlines. | Degrees | Average | EarthEnv-DEM90 (Robinson et al., 2014) |
| Stream gradient average | The ratio between the elevation drop and the length of the river reach. | Decimetres per kilometre | Average | EarthEnv-DEM90 (Robinson et al., 2014) |
| Precipitation annual | Long-term (1950-2000) precipitation within the basin. | Millimetres | Annual average | WorldClim v1.4 (Hijmans et al., 2005) |
| Aridity Index | Ranges from 0 to 1: a value of 0 represents areas with no precipitation and 1 represent areas where P >= PET. | Index value | Average | Global Aridity Index (Zomer et al., 2008) |
| Climate moisture index | Ranges from -1 to 1: wet climates yield positive values and dry climates yield negative values. | Index value | Average | WorldClim & Global-PET (Hijmans et al., 2005) |
| Snow cover annual | Represents the period 2002-2015. | Percent cover | Annual average | MODIS/Aqua (Hall and Riggs, 2016) |
| Forest cover extent | Combines the land classes 1 to 8 in the GLC2000 land cover map. | Percent cover | Spatial extent | GLC2000 (Bartholomé and Belward, 2005) |
| Cropland extent | Represents the year 2000. | Percent cover | Spatial extent | EarthStat (Ramankutty et al., 2008) |
| Pasture extent | Represents the year 2000. | Percent cover | Spatial extent | EarthStat (Ramankutty et al., 2008) |
| Irrigated area extent | Area equipped for irrigation the year 2005. | Percent cover | Spatial extent | HID v1.0 (Siebert et al., 2015) |
| Glacier extent | Based on a global glacier inventory between 1950 and 2015. | Percent cover | Spatial extent | GLIMS (Raup et al. 2007) |
| Permafrost extent | Modelled occurrence of permafrost using estimates for the period 1961-1990. | Percent cover | Spatial extent | PZI (Gruber, 2012) |
| Population count 2010 | Population count for the year 2010 within the basin and the upstream watershed respectively. | Count (thousands) | Sum | GPW v4 (CIESIN, 2016a) |
| Population density 2010 | Population density for the year 2010 within the basin and the upstream watershed respectively. | People per square kilometre | Average | GPW v4 (CIESIN, 2016b) |
| Urban extent | Combining the low-density clusters (class 2) and high-density clusters (class 3) in the settlement model grid GHS-SMOD for the year 2015. | Percent cover | Spatial extent | GHS S-MOD v1.0 (Pesaresi and Freire, 2016) |
| GDP sum 2015 | The GDP total for the year 2015, within the basin and upstream watershed respectively. | U.S. dollars | Sum | GDP PPP v2 (Kummu et al., 2018) |




## 3.2 Country selection

The subsequent exposure and area analyses have been conducted for 26 countries in the Global South: Bangladesh, Bolivia, Brazil, Cambodia, Central African Republic, Colombia, Republic of the Congo, Ecuador, Ghana, Guatemala, Honduras, India, Indonesia, Kenya, Lao People's Democratic Republic, Liberia, Malawi, Mozambique, Nicaragua, Nigeria, Peru, Sri Lanka,

Thailand, Uganda, United Republic of Tanzania, and Viet Nam. These countries are situated in the Eastern, Middle and Western parts of Africa, South-Eastern Asia, Southern Asia, Central America and South America.

The countries were selected based on four criteria. First, we only included countries fully covered by all six datasets used for the comparative exposure analysis (GFPLAIN, JRC, GAR, GHS, HRSL and WorldPop). Second, only countries belonging to the low-, lower middle- and upper middle-income groups were selected (Figure 1), here classified using GNI per capita for the

year 2015 (The World Bank, 2021b). The assumption was that, in general, the high-income countries have less need for global flood products since they tend to be more well-equipped with national flood maps. Third, the selected countries all have a relatively high degree of riverine flood risk (Figure 1), here represented by the combination of the population within the 100-year flood hazard map (intersecting GHS and JRC) and actual disaster reports of people affected by riverine floods 1900-2020 (Guha-Sapir et al., 2014). Finally, we excluded countries with the main climate classification being arid or snowy according

to the World Map of Köppen-Geiger (Beck et al., 2018).

## 3.3 Exposure and area analysis

The number of people located in flood-prone zones was calculated for each country by intersecting the binary and masked images of GFPLAIN, JRC-500, JRC-100, GAR-500, and GAR-100 with the population maps of GHS, HRSL and WorldPop. Country boundaries were outlined with The Global Administrative Unit Layers (GAUL) dataset for the year 2015 (FAO UN,

2015). The number of people living in flood-prone zones was then exported for all countries and dataset combinations. The population maps were also used for calculating country population totals, which have been compared to numbers provided by the World Bank in Table A6. The percentage of the country population living in flood-prone zones were then derived for each country and dataset combination.

For all area calculations of raster images, we used a pixel area function in Google Earth Engine (Gorelick et al., 2017) that

minimizes projection distortions, which otherwise may be an issue when calculating areas over large regions. The function first calculated the area of each pixel, using individual Lambert Azimuthal Equal Area projections for each block of pixels. The resulting grid of pixel areas was subsequently multiplied with the masked and binary images, for which the sum could be exported for each region of interest.


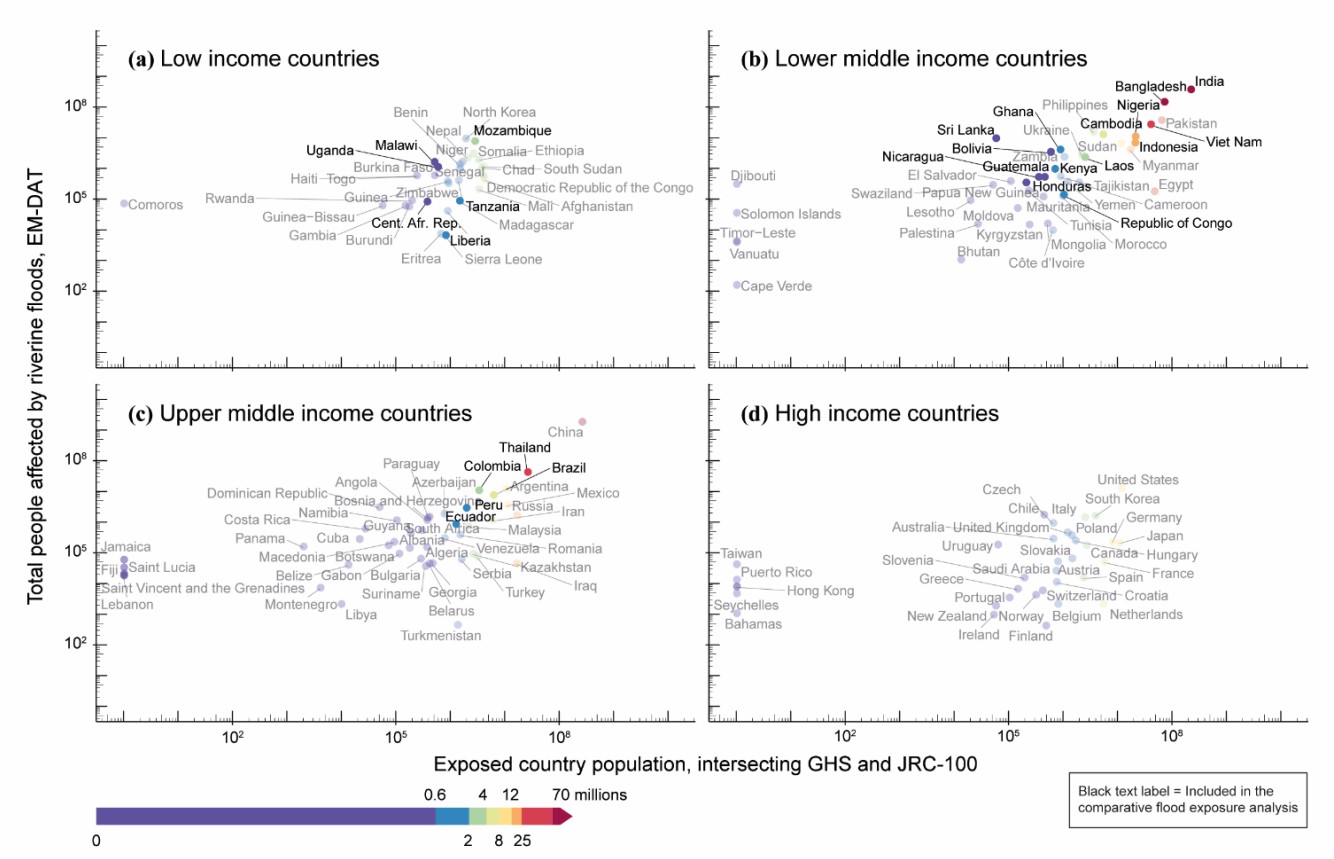

**Figure 1: Flood risk in the 26 countries for which we conducted the comparative riverine flood exposure analysis, here marked with black text labels. Flood risk is here represented as the combination of the country population within the 100-year flood hazard map (intersecting the Global Human Settlement Population count for 2015 with the JRC flood hazard map) and actual disaster reports of people affected by riverine floods 1900-2020 as included in the Emergency Events Database EM-DAT. Please note that the axes have logarithmic scales. The country income classification is based on GNI per capita for the year 2015, given by the World Bank.**

The total land surface area was calculated for all 26 countries by subtracting the total country area, as outlined by the polygons in the GAUL dataset (FAO UN, 2015), with the area of permanent surface water as given by the raster water mask MOD44W (Carroll et al., 2009). The area estimations of flood-prone zones, given by the binary and masked images of GFPLAIN, JRC-500, JRC-100, GAR-500 and GAR-100, were finally exported for all countries. The percentage of country area that is flood-prone was then derived for all countries and flood maps.



## 4 Results and discussions

In this section, we first discuss the model agreement between the flood maps in Sect. 4.1 and then turn to the implications on riverine exposure analysis in Sect. 4.2.

### 4.1 Model agreement of flood-prone zones

### 4.1.1 Model agreement across model combinations

The overall spatial pattern of the floodplain map GFPLAIN resembles the flood hazard maps of GAR more than the flood hazard maps of JRC (Figure 2). A possible explanation of this is that GFPLAIN and GAR cover larger parts of each river network compared to JRC since they exhibit lower inclusive thresholds for the upstream drainage area. Figure 2 shows that
the model agreement between the floodplain map GFPLAIN and the flood hazard maps of GAR is, on average, around 35 %. The corresponding agreement between GFPLAIN and the flood hazard maps of JRC is around 30 %. These agreement levels are in line with the results of Trigg et al. (2016) when comparing the outputs of six global flood models for the African continent, finding an average model agreement around 30 to 40 %. It is also clear from Figure 2 that there is a large spread of agreement scores across all 2776 river basins: all pairwise comparisons range between the maximum model agreement value
100 % and very close to the minimum value of 0 %.

The pairwise model agreement evaluation also confirms that the model agreement between GFPLAIN and the hazard maps is higher for the return period of 500-years compared to 100-years. This is consistent with previous findings in the literature about how the morphology of rivers, which the hydrogeomorphic floodplain maps delineate, is predominantly shaped by high-magnitude, low-recurrence flood events (Annis et al., 2019; Bhowmik, 1984; Nardi et al., 2018). Figure 2 pinpoints, however,
that choice of hazard model has a greater influence on the degree of model agreement with GFPLAIN, compared to the choice of return period.

Contrary to expectations, the model agreement between the hazard maps of JRC and GAR is lower compared to their agreement with GFPLAIN. The median agreement values across all river basins are found to be 0.34 for GFPLAIN and GAR-500, 0.27 for GFPLAIN and JRC-500 and 0.20 for GAR-500 and JRC-500. The reasons for this result are not evident, but it supports
earlier findings of significant disagreement among individual global flood models (Trigg et al., 2016; Ward et al., 2015). All distributions from the pairwise model agreement evaluation show positive skewness, meaning that the bulk of river basins have low values (Figure 2). The portion of river basins having low agreement scores is highest for the GAR and JRC comparison, illustrated by the relatively big differences between the average and median values. A Kruskal-Wallis test showed that there is a significant difference in model agreement between the distributions, $H(5) = 1217$, $p < 0.001$, the pairwise
comparisons from the post hoc test can be found in Table A2.


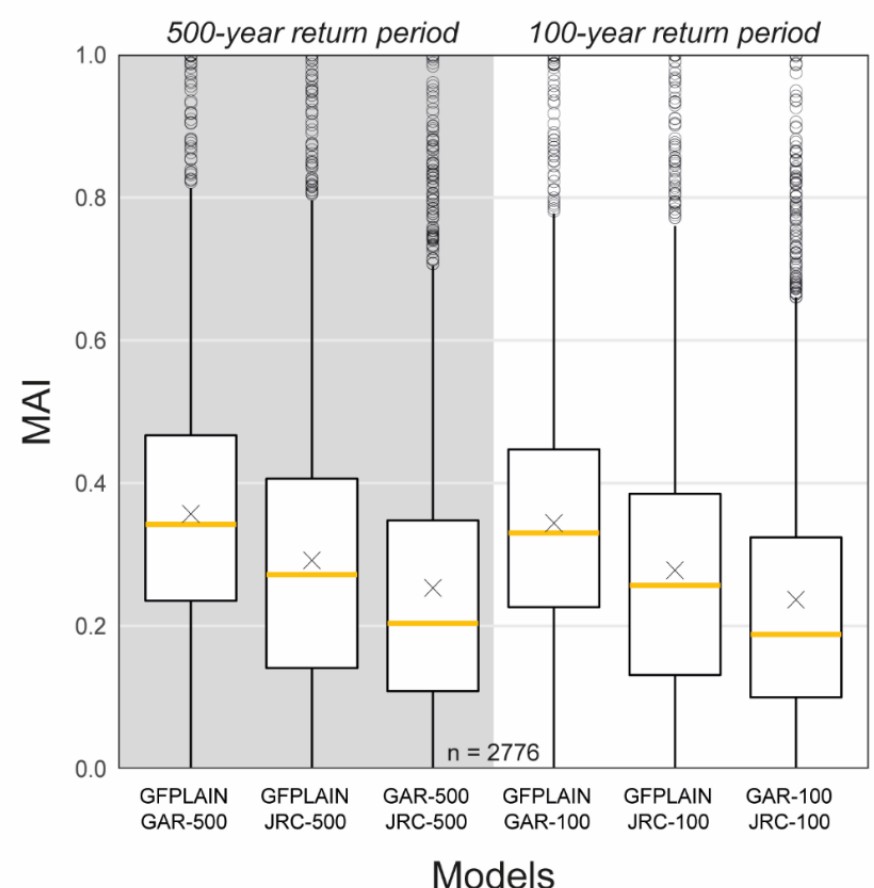

**Figure 2 Pairwise evaluation of model agreement between the hydrogeomorphic floodplain map GFPLAIN and the flood hazard maps GAR and JRC, with 100- and 500-year return period. A higher return period shows higher agreement with GFPLAIN, but the choice of flood model influences the level of agreement even more. The flood hazard maps GAR and JRC show the lowest level of agreement. The index has here been calculated for the 2776 river basins across the world that are covered by all three models.**

### 4.1.2 Model agreement across geographic conditions

We also analysed how MAI-500, the model agreement between GFPLAIN, GAR-500 and JRC-500, varies across space and a set of hydro-environmental groups. We chose to represent the model agreement with MAI-500 since we generally found a higher model agreement using the 500-year return period, compared to the 100-year return period. Figure 3 provides the spatial distribution of MAI-500 across all 2776 river basins, and local clusters of high and low model agreement basins as identified from the spatial autocorrelation analysis. These maps suggest that the model agreement is associated with factors related to climate, topography, and coastal proximity.

**Figure 3: (a) Spatial distribution of MAI-500, the model agreement between the hydrogeomorphic floodplain map GFPLAIN and the flood hazard maps GAR and JRC with a 500-year return period. The model agreement index ranges between 0 % (no agreement) and 100 % (full agreement) and was here calculated for all the river basins of the world that are covered by all three models. (b) Local clusters of river basins with high or low agreement values relative to the mean, identified from the spatial autocorrelation analysis. Poor model agreement is particularly found in dry climate regions, as exemplified by the snow and ice regions in North America and Asia, and the desert regions in Africa. The highest agreement is generally found in humid regions.**

The river basins with the highest model agreement are generally located in humid regions (Figure 3). For instance, it is the equatorial humid climate regions that exhibit the highest model agreement in the South American continent, including the north-western part of Brazil and the north-eastern part of Peru. The high agreement cluster in North America, near the Mississippi River Delta, as well as the high agreement cluster in Southeast Europe, are also situated in warm temperate and





fully humid regions. On the contrary, a poor model agreement is generally found in dry regions. The snow and ice regions in North America entail large basin clusters with a low model agreement. Arid desert regions also exhibit low model agreement clusters, such as the Kalahari Desert in Southern Africa.

Relating to topography, Figure 3 also shows that alpine regions like the Andes in South America and High-mountain Asia exhibit poor model agreement. This can, at least partly, be explained by the same regions being dry in the sense that they are

snow-covered. It also seems possible that very steep regions would be prone to error. However, very flat regions also tend to exhibit poor model agreement, across climate types. This may be exemplified by the flat grassland of the humid region Pampa in Argentina, and the flat inland areas in the tropical islands of Indonesia.

Figure 3a furthermore illustrates that coastal river basins seem to exhibit low model agreement, visible as white lines along many coastlines. This tendency is not captured well by the clusters in Figure 3b since one single low agreement basin near the

coast would not count as a cluster. A possible explanation for the low agreement in coastal river basins might be that the individual riverine flood maps differ in how they mask coastal areas. For instance, GFPLAIN tends to mask areas near the coast, while JRC does not. Another possible explanation may be due to complex flood hydraulics, in line with previous findings of low agreement in deltas (Trigg et al., 2016). However, it can also be seen in Figure 3 that major river deltas often exhibit high model agreement on basin level, as exemplified by the Mississippi River Delta in the U.S. and the Mekong River Delta

in Viet Nam and Cambodia. This is an indication that the volume of the river also affects the model agreement, which seems possible since large river discharge imposes large forces on the surrounding landforms, the very shapes that GFPLAIN aims to capture.

Thus far, we have elaborated on what might cause the spatial variation of MAI-500 based on a visual inspection of the global maps in Figure 3. We also quantified these results by assessing how the MAI-500 value varies between a set of factoring

groups: geographic region, river stream order, and freshwater major habitat type. Across the basins situated on each geographic region, Figure 4a conveys that the highest model agreement can, on average, be found in South America (39 %) and Asia (39 %), while North America (29 %) and Africa (32 %) exhibit the lowest model agreement. This supports, again, that humid regions tend to perform better compared to dry regions. The Kruskal-Wallis test showed that there is a significant difference in model agreement between the regions, $H(6) = 146$, $p < 0.001$, the pairwise comparisons from the post hoc test can be found

in Table A3.



**Figure 4: How the model agreement between GFPLAIN, GAR-500 and JRC-500 varies with (a) geographic region, (b) river stream order, and (c) freshwater major habitat type. The model agreement index ranges between 0 % (no agreement) and 100 % (full agreement) and was here calculated for all the river basins of the world that are covered by all three models. The number of river**

**basins corresponding to each group is indicated in numbers and box width. Coastal river basins tend to have lower model agreement scores, compared to inland river basins. The freshwater major habitat type specifies the spatially dominating habitat type in each river basin (WWF and TNC, 2019). The habitat type "Large river deltas" only includes regions that exhibit both deltaic physical features and deltaic fauna. The habitat type "Large lakes" includes river basins that are dominated by large lentic systems.**





Figure 4b illustrates how MAI-500 varies with river stream order, supporting that model agreement tends to be negatively
associated with coastal proximity. Furthermore, the Kruskal-Wallis test showed that the model agreement significantly varies
between river stream orders, $H(3) = 101$, $p < 0.001$. More specifically, the model agreement in coastal basins, stream order 0,
was identified as significantly lower compared to inland stream order groups, verified by the post hoc Wilcoxon rank test (p-
values $< 0.001$, see Table A4).

We now move on to the freshwater major habitat types, a classification based on the spatially dominating habitat type in each
river basin (WWF and TNC, 2019). Before presenting the model agreement variations between the habitat types, however, we
point out that a classification on river basin level will inevitably contain multiple smaller habitat types. For instance, the habitat
type "Large lakes" includes river basins that are dominated by large lentic systems, in our case covering three large regions:
Lake Baikal in Siberia, Lake Malawi in Africa and Michigan-Huron in North America. But a river basin of this habitat type
can, for example, also contain grassy savannas or swamps (WWF and TNC, 2019). Another point is that the habitat type "Large
river deltas" only include regions that exhibit both deltaic physical features and deltaic fauna (WWF and TNC, 2019). This
habitat type covers four large regions within our analysed river basins: the Niger Delta in Nigeria, the Mekong Delta in
Viet Nam and Cambodia, the Orinoco Delta in Venezuela, and Brazilian delta regions between the rivers Amazon and Mearim.
In other words, deltas without the characteristic deltaic fauna do not belong to this habitat type. For instance, the river basins
containing the Mississippi River Delta belong to the habitat type "Temperate floodplain rivers and wetland complexes".
Another point is that the difference between rivers classified as floodplain rivers and upland rivers is the presence of cyclically
flooded floodplains. Floodplain rivers are characterised by having cyclically flooded floodplains (today or historically), while
the upland rivers are not (WWF and TNC, 2019). Upland rivers can for instance be tributaries of large river systems.

Figure 4c shows how the model agreement varies between the freshwater major habitat types. Floodplain rivers, in temperate
and tropical climate regions, exhibit higher model agreement compared to upland rivers, followed by coastal rivers. We can
also see, as previously discussed, that freshwaters in arid, here expressed as xeric, and montane regions exhibit poor model
agreement. The group "Large river deltas" is ranked as having the highest model agreement, clearly influenced by the high
agreement in large river deltas, like the Mekong. These results should be interpreted with caution, however, due to the
unbalanced sample sizes between the groups (Figure 4c). Nonetheless, the inherent order between the groups supports previous
discussion about how factors related to climate, topography, coastal proximity and river volume affect MAI-500 values. The
Kruskal-Wallis test showed that the agreement difference is significant between the habitat types, $H(10) = 478$, $p < 0.001$. The
pairwise comparisons from the post hoc test can be found in Table A5.




### 4.1.3 Model agreement association with hydro-environmental attributes

Section 4.1 has so far demonstrated that MAI-500, the level of model agreement between GFPLAIN, GAR-500 and JRC-500, is linked to factors related to climate, topography, coastal proximity and river volume. We have done this by visually exploring
the spatial pattern of the model agreement, and by grouping the values according to geographic region, river stream order and freshwater major habitat type. Each of these habitat groups contains a variety of geographic characteristics, so we also analysed the level of association between model agreement and individual hydro-environmental attributes and investigated whether the association differs between the model agreement pairs.

Figure 5 presents the Spearman rank-order correlation coefficients in a correlogram, where the variables have been ordered
according to a hierarchical clustering method. The correlogram in Figure 5 supports the statement that wet regions tend to exhibit higher model agreement compared to dry regions. Precipitation, land surface runoff, the aridity index and the climate moisture index positively associate with the level of model agreement. Precipitation exhibit the strongest positive association among these variables, especially for $MAI_{\text{GFPLAIN JRC-500}}$. The association levels between $MAI_{\text{GAR-500 JRC-500}}$ and these climatic variables are, however, relatively weak or even insignificant (the aridity index and the climate moisture index). In other words,
the level of agreement between GFPLAIN and the hazard maps seems to be positively influenced by a wet climate, while the agreement between JRC and GAR seems to be controlled by other factors. This result further supports the comment that the skill of hydrogeomorphic floodplain maps to represent flood hazard depends on the availability of water.

The variables that represent the magnitude of river discharge also exhibit a positive association with the model agreement, in an even stronger sense compared to the previously mentioned climatic variables (Figure 5). This is particularly evident for the
variables natural discharge and upstream river volume, having moderate to strong positive association with all model agreement distributions. For the very same reason, we can see that the size of the upstream drainage area exhibit a positive association with the model agreement, but to a moderate degree. As previously discussed, increased discharge means larger forces forming the landscape, and hence more well-defined floodplains. The size of the river basin being evaluated, on the other hand, co-variates with the model agreement in a weak negative direction. One explanation for this is that very small river
basins get maximum model agreement scores when they are fully covered by all three flood models.

The association between model agreement and the variables elevation, slope and stream gradient is significant in a negative direction (Figure 5). This finding supports previous discussion about how high altitude and steep regions tend to exhibit poor model agreement. The association between these topographic variables is strongest for the model agreement between GFPLAIN and GAR, whereas the association is weakest for the agreement between GFPLAIN and JRC. The reason for this
result is not evident, but one explanation could be that the one-dimensional hydraulic modelling of GAR tends to be sensitive to these topographic variations. These findings may also be somewhat limited since the relationship between model agreement and, for instance, the slope is not monotonic: we have already observed how many very flat areas also exhibit poor model agreement (Figure 3).



**Figure 5 Spearman rank-order correlation coefficients between the flood model agreement scores and 27 hydro-environmental attributes. Wetter conditions due to climate (precipitation, land surface runoff) and river size (e.g. river volume, natural discharge) are positively associated with the model agreement between the flood maps. Topographic variables like high altitude (elevation) and steep slope (e.g. slope and stream gradient) are negatively associated with the model agreement, especially between GFPLAIN and GAR-500. The correlogram is using a hierarchical clustering order. Please note that the Aridity index increases with humidity.**

The overall association between model agreement and the anthropogenic influences (population count, GDP) and the land cover characteristics (e.g. urban extent, cropland extent) is weak. These results are likely to be related to the scale of the analysis, any possible association would be averaged out on the aggregated basin level. Nardi et al. (2018) predominantly found consistencies between a hydrogeomorphic floodplain map and flood hazard maps in areas unaltered by humans.



## 4.2 Implications on exposure analysis

Our analysis has shown that the level of agreement between GFPLAIN and the flood hazard maps GAR and JRC is, on average, higher compared to the agreement between GAR and JRC. Furthermore, we have shown that the model agreement is linked to factors related to climate, topography, river size and coastal proximity. We further analysed how these differences affect riverine flood exposure analysis, which we have conducted for 26 countries using the population maps GHS, WorldPop and HRSL.

The flood hazard maps GAR-500 and GAR-100 cover an area of ~3.3 M km$^2$ and ~3.18 M km$^2$ across the 26 countries, while the floodplain map GFPLAIN covers ~3.17 M km$^2$. The hazard maps JRC-500 and JRC-100 cover an area of ~1.77 M km$^2$ and ~1.63 M km$^2$ respectively. One explanation for this large difference is that GAR and GFPLAIN include smaller rivers compared to JRC. Figure 6 shows the flood exposure estimates of each map combination, aggregated across all 26 countries. Interestingly, GFPLAIN yields a larger exposure estimate compared to GAR and JRC, even though GAR-500 covers a larger total area than GFPLAIN. One explanation of the higher population estimate given by GFPLAIN is that the floodplain map covers a larger area of densely populated regions in India, resulting in 44 M more exposed people with GFPLAIN compared to GAR-500 (using the population map HRSL). This difference is, however, counteracted by the instances where GAR-500 yields larger exposure estimates, for example, 21 M more exposed people in Bangladesh.

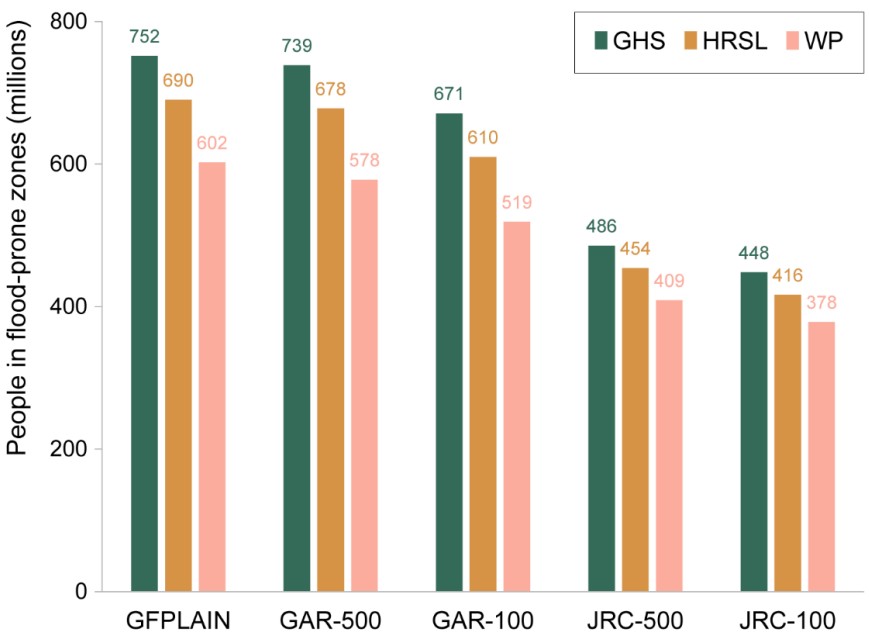

**Figure 6 Total exposed population (millions) across all 26 countries, comparing the hydrogeomorphic floodplain map GFPLAIN with the flood hazard maps GAR and JRC with 100- and 500-year return periods. The population estimates are given by the Global Human Settlement Layer, the High Resolution Settlement Layer and WorldPop. The floodplain map GFPLAIN yields the highest exposure estimates, even though GAR-500 covers a slightly larger total area. The choice of flood and population maps affect the results more than the choice of the return period.**



We would like to highlight that 1) the choice of flood model influences the exposure estimates to a higher degree than the choice of the hazard return period. We think that this finding provides an important perspective to end-users of global flood maps; 2) we observe an inherent order across the population maps (GHS > HRSL > WorldPop), which can be explained by the varying population totals across these datasets (Table A6). WorldPop, for instance, exhibits considerably lower population

values for all 26 countries. Scaling the country totals would enable a more consistent comparison of the exposure hit between the individual population datasets. We did not scale the population estimates, however, since this study aims to capture how dataset choice affects exposure estimates.

Let us now turn to discuss how the riverine exposure estimates vary across individual countries and map combinations. Figure 7 presents the flood-prone area and exposed population as proportions for each country. Among our analysed 26 countries,

countries in Southern and South-East Asia stand out as having the highest riverine flood exposure. For instance, the largest number of people living within riverine flood-prone zones can be found in India (346 M), Bangladesh (100 M), Viet Nam (53 M) and Indonesia (37 M). The proportion of exposed people is largest for Bangladesh (59 %), Viet Nam (54 %), Cambodia (53 %) and Laos (50 %). The proportion of land area that is flood-prone is largest for Bangladesh (57 %), Cambodia (28 %), Bolivia (26 %) and Thailand (21 %). The numbers listed here represent the intersection of GFPLAIN and HRSL. All results

of the area calculations and exposure estimations have been made available in Table A7, Table A8 and Table A9.

It can also be seen in Figure 7 that GAR-500 covers the largest area in 17 countries, whereas GFPLAIN covers the largest area in the remaining 9 countries. Honduras is the only country where JRC covers a larger area compared to GFPLAIN, also resulting in larger exposure estimates. This is explained by the fact that coastal areas make up a large part of this country, and these are more often masked from GFPLAIN compared to JRC. This tendency is not consistent, however, and for instance,

does not apply to Sri Lanka. One of the issues emerging from this finding is that masked areas near the coast can affect the exposure analysis result. Even though one would not expect the usage of riverine flood maps for estimating coastal flood risk, it might still affect riverine exposure estimations on, say, country level, since the coastal areas also tend to be densely populated. This is most certainly the case in Liberia. GFPLAIN covers a larger area compared to JRC, but misses parts of the densely populated coastal city Monrovia, resulting in a 0.43 M smaller exposure estimate compared to JRC. At the same time,

variations among the flood maps can also affect the coverage of inland cities, as in the case of Colombia. GAR-500 covers a larger area compared to GFPLAIN, but GFPLAIN yields a 4.5 M larger exposure estimate from covering a larger part of the capital city Bogotá.

As far as the influence of the return period is concerned, the difference between the exposure estimates of JRC-500 and JRC-100 is largely proportional to the corresponding area difference. This is also the case for the hazard maps of GAR, with some

exceptions. The area ratio between GAR-500 and GAR-100 in Thailand, for example, is 1.09, while the corresponding exposure ratio using HRSL is 1.83. Similar results can be found for Indonesia (1.04 compared to 1.22), Brazil (1.03 compared to 1.17) and Colombia (1.02 compared to 1.13). This finding might reveal how populations in these countries tend to live outside, but close to, the 100-year flood hazard zone.







**Figure 7** Riverine flood exposure estimates, aggregated at country level, using the hydrogeomorphic floodplain map GFPLAIN and the flood hazard maps of JRC and GAR with a 500-year return period. The green, orange and pink bars indicate the percentage country population in flood-prone zones using the population maps Global Human Settlement, High Resolution Settlement Layer and WorldPop respectively. The blue bars indicate the percentage of land area that is flood-prone. Country boundaries are outlined by the GAUL 2015 dataset and permanent water is masked by the product MOD44W.





Figure 8 illustrates some of the implications that the dataset differences have on riverine exposure analysis. Figure 8a covers a part of the Amazon River in Brazil and was previously, in Sect. 4.1.2, identified as a high model agreement cluster. The spatial pattern of GFPLAIN, JRC-500 and GAR-500 are quite similar for this major river. GFPLAIN and GAR cover smaller river tributaries compared to JRC, but this difference does not affect the exposure estimates since the population is clustered in the city of Manaus near the main river. Figure 8b illustrates a major discrepancy in handling of coastal regions between the

individual models, exemplified by the Niger Delta in Nigeria. JRC covers the entire river basin, while both GFPLAIN and GAR have masked the coastal areas to different degrees. These discrepancies affect the exposure hits in cities like Port Harcourt and Warri.

Figure 8c covers a part of the Krishna River in India. This map also illustrates how GFPLAIN and GAR include smaller rivers than JRC, as the case for the river tributaries north of the city Khammam. We can also see how GFPLAIN has masked the

coastal area in the southeast corner. The spatial pattern of GFPLAIN reaches the city of Guntur, nonetheless, in contrast to GAR and JRC. Figure 8d covers a part of the Mekong River in Cambodia, also identified as a high model agreement cluster in Sect. 4.1.2. The spatial pattern is indeed quite similar between the flood maps for this large river, but with some discrepancy. For instance, GFPLAIN does not exhibit the same gaps in the flood extent, as can be seen in the north part of the JRC and GAR flood maps. These gaps represent hilly areas that in reality would be unlikely to flood, and could be an artefact of the

hydrogeomorphic delineation method. GFPLAIN furthermore covers a larger portion of the capital Phnom Penh, affecting the exposure hits.

In closing this section, we want to pinpoint one last remark from the findings illustrated by Figure 7 related to the inherent order between the population maps (GHS > HRSL > WorldPop). As discussed, a scaling of the country totals would enable a more consistent comparison of the exposure hit between the individual population datasets. Nonetheless, we have indeed

conducted a form of scaling in Figure 7 when normalizing the exposed population estimates with the population totals. The proportion of the exposed population is still generally highest for the population map GHS, whereas the results vary for HRSL and WorldPop. This can be contrasted to the study of Smith et al. (2019), finding considerably lower exposure estimates for HRSL, compared to GHS and WorldPop, when scaling the demographic datasets to share the country totals. But an important factor of their finding, besides the scaling of country totals, is that the hazard data needs to be high resolution to make use of

the detailed settlement representation of HRSL (Smith et al., 2019). We think that this also highlights an important aspect of the usability of hydrogeomorphic floodplain maps in riverine exposure analysis. The opportunity to delineate high-resolution floodplain maps from new terrain models can play an important role in conducting riverine flood exposure estimations in data-poor regions, if one wishes to make the best use of the detailed representations of new population maps, like HRSL.


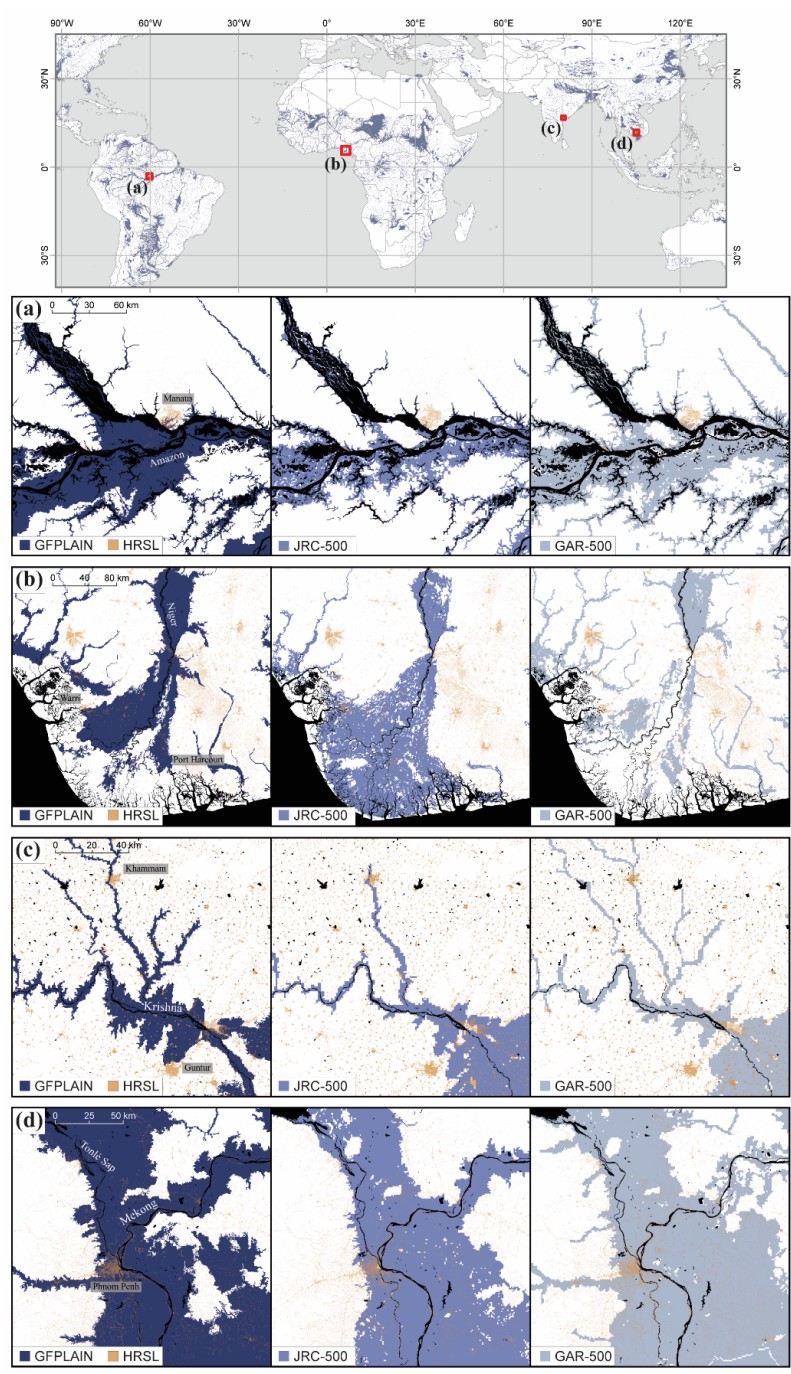

**Figure 8 Spatial patterns of the hydrogeomorphic floodplain map GFPLAIN compared to the flood hazard maps JRC and GAR with a 500-year return period, for the rivers (a) Amazon in Brazil, (b) Niger in Nigeria, (c) Krishna in India, and (d) the Mekong in Cambodia. The brown pixels indicate settlements mapped with the High Resolution Settlement Layer. The black pixels indicate the permanent water mask. GFPLAIN and GAR cover more rivers compared to JRC due to having lower thresholds of upstream drainage area. There is a major discrepancy between the treatment of coastal regions between the individual models, here particularly evident for the Niger Delta (b).**




## 5. Conclusions

In this paper, we evaluated model agreement between the hydrogeomorphic floodplain map GFPLAIN with the flood hazard maps of JRC and GAR across geographic conditions and a set of hydro-environmental attributes. We demonstrated that the level of model agreement between GFPLAIN and the flood hazard maps is linked to climatic conditions, topography, coastal proximity and river volume. We also conducted riverine exposure analysis for 26 countries by intersecting the flood maps with three demographic datasets to explore how model differences affect riverine flood exposure estimations. Our findings can be

summarized by three main points.

First, our results confirm that the consistency between GFPLAIN and the flood hazard maps increases with the return period. The choice of model for the flood hazard map is, nevertheless, more important for evaluating model agreement on river basin level and, also, for affecting the results of exposure analysis.

Second, our study confirms that the results of riverine exposure analysis are highly dependent upon the choice of datasets.

Contrary to expectations, the model agreement between the JRC and GAR hazard maps is lower compared to their agreement with GFPLAIN. The median agreement values across all river basins are found to be 0.34 for GFPLAIN and GAR-500, 0.27 for GFPLAIN and JRC-500 and 0.20 for GAR-500 and JRC-500. There is a large spread across all basins, however, ranging between the maximum model agreement value 1 and very close to the minimum value 0. This finding (yet again) stresses the uncertainties of global flood models.

Third, the agreement level between GFPLAIN and the flood hazard maps suggests that hydrogeomorphic terrain analysis can indeed be viewed as a valuable way of estimating flood-prone zones, especially in data-poor regions. The highest model agreement was generally found for large rivers in temperate or tropical climate regions. However, we do not wish to reduce hydrogeomorphic methods as mere substitutes for global flood models: the individual methodologies ultimately serve different purposes. Furthermore, floodplain maps built by hydrogeomorphic terrain analysis should be used with caution in regions that

are dry, steep, very flat or near the coast. The tendency by GFPLAIN to not cover coastal areas may in many cases affect the riverine flood exposure estimates, even on country level, since coastal areas also tend to be densely populated.

Inter-model comparisons like this study do not answer the question of how well the individual flood layers agree with actual flood events. Future research can build upon this comparative study by conducting a validation analysis using flood delineation from satellite imagery, similar to the recent work by Bernhofen et al. (2018). Future studies can also investigate how model

agreement varies with the fluvial geomorphology of the rivers (e.g. as given by the global river classification dataset GloRiC (Ouellet Dallaire et al., 2019)) or land cover (e.g. as given by the ESA CCI 2015 Land Cover Map (Defourny et al., 2017)).

To conclude, this study provided initial insights on how and where hydrogeomorphic floodplain maps could serve as candidates for identifying flood-prone areas in riverine flood risk studies. One particular benefit of hydrogeomorphic terrain analysis is that it does not require hydrological information to produce high-resolution floodplain maps whenever refined terrain models

become available. This could particularly play an important role in riverine flood exposure analysis, as the detail of the hazard





layer needs to meet the detail of the settlement layer to avoid overestimations of flood exposure, which is especially important in dispersed rural regions.

## Appendix A

**Table A1: A technical summary of the datasets used for riverine flood exposure analysis.**

| Dataset | Credit | Variable | Format | Input data description | Model description |
|---|---|---|---|---|---|
| GFPLAIN250m **(GFPLAIN)** | (Nardi et al., 2019) | Binary map of river floodplains. | GeoTIFF, 8.33 arcsec (~250 m), 60° N, 56° S | Digital Elevation Model SRTM version 4.1. | Identifies alluvium extent with geomorphic terrain analysis. Only includes river basins with a contributing area > 1000 km². |
| Flood Hazard Map of the World **(JRC)** | (Dottori et al., 2016) | Pixel values indicate maximum water depth (m) of flood-prone areas for flood event with the corresponding return period. | GeoTIFF, 30 arcsec (~1 km), Global | ERA-Interim meteorological data 1980-2013. HydroSHEDS (based on SRTM), digital elevation model GTOPO30, surface roughness values from the Global Land Cover 2000 map, river width from the Global River Width Database. | Uses hydrological simulations of GloFAS to derive river discharge. Gumbel extreme value distribution is used for fitting the daily annual discharge maxima to peak discharge maps for the respective return periods, for all river basins with an upstream drainage area > 5000 km² and river width >100 m. Streamflow is downscaled on the river network HydroSHEDS and used as input for local flood inundation simulations with a 2-D hydraulic model CA2D. |
| Global Assessment Report on Disaster Risk Reduction 2015 flood hazard maps **(GAR)** | (CIMA Foundation, 2015) | Pixel values indicate maximum water depth (cm) of flood-prone areas for flood event with the corresponding return period. | GeoTIFF, 30 arcsec (~1 km), 60° N, 56° S. | River discharge station data (e.g. GRDC, RivDIS, GHCDN) covering > 8000 stations. The Global Reservoir and Dam Database (GRanD). SRTM version 2, SRTM water body data, HydroSHEDS. Land cover data (GLC 2000, ESA, GLWD) for specifying basin characteristics for statistical analysis, and estimate surface roughness. Climate datasets (CRU TS and CHIRPS) for specifying basin characteristics for statistical analysis. | Uses statistical regionalization techniques on river discharge measurements to compute extreme discharge values, which are used as input to a one-dimensional hydraulic flood inundation model. Only includes river basins with a contributing area > 1000 km². The native model resolution is 3 arcsec (~ 90 m), the hazard maps have subsequently been resampled to 30 arcsec. |
| High Resolution Settlement Layer **(HRSL)** | (Facebook Connectivity Lab and CIESIN, 2016) | Pixel values indicate the number of inhabitants for the year 2018. | GeoTIFF, 1 arcsec (~30 m), available for >140 countries. | Population census data, satellite imagery of Digital Globe (0.5 m). | Uses convolutional neural networks to detect buildings from high-resolution satellite data. Population estimates from census data are allocated to the buildings via a binary dasymetric modelling approach. |
| Global Human Settlement Layer - Population Grid 2015, version R2019A **(GHS)** | (Schiavina et al., 2019) | Pixel values indicate the number of inhabitants for the year 2015. | GeoTIFF, 9 arcsec (~250 m), Global | Population estimates of CIESIN Gridded Population of the World version 4.10. Built-up land of GHSL, detected by satellite imagery of Landsat (30 m). The population estimates are disaggregated from census or administrative units through the distribution and density of built-up land. | Uses a binary dasymetric modelling approach for allocating subnational census data to built-up areas. |
| WorldPop Estimated Residential Population per 100x100m Grid Square **(WorldPop)** | (Gaughan et al., 2013; Linard et al., 2012; Sorichetta et al., 2015; Tatem, 2017) | Pixel values indicate the number of inhabitants for the year 2018. | GeoTIFF, 3 arcsec (~90 m), Global | Uses a range of input datasets, e.g. land cover, roads, slope, nighttime lights. Built-up is detected by satellite imagery of Landsat (30 m). | Uses a random forests-based dasymetric modelling approach for allocating census counts to built-up areas. |




**Table A2: Summary statistics and pairwise comparisons of the model agreement between each pair of flood maps. A Kruskal-Wallis test showed that there is a significant difference in model agreement between groups, H(5) =1217, p < 0.001. The pairwise comparisons, for detecting significant differences between the groups, have been conducted using the Wilcoxon rank-sum test with continuity correction.**

| | n | Mean | S.D. | (1) | (2) | (3) | (4) | (5) |
|---|---|---|---|---|---|---|---|---|
| 1. MAI$_{GFPLAIN\ GAR-500}$ | 2776 | 0.357 | 0.178 | - | | | | |
| 2. MAI$_{GFPLAIN\ JRC-500}$ | 2776 | 0.292 | 0.199 | *** | - | | | |
| 3. MAI$_{GAR-500\ JRC-500}$ | 2776 | 0.253 | 0.194 | *** | *** | - | | |
| 4. MAI$_{GFPLAIN\ GAR-100}$ | 2776 | 0.344 | 0.174 | ** | *** | *** | - | |
| 5. MAI$_{GFPLAIN\ JRC-100}$ | 2776 | 0.278 | 0.194 | *** | ** | *** | *** | - |
| 6. MAI$_{GAR-100\ JRC-100}$ | 2776 | 0.237 | 0.187 | *** | *** | *** | *** | *** |

Note. *p < .05, **p < .01, ***p < .001


**Table A3: Summary statistics and pairwise comparisons of how MAI-500 (the model agreement index of GFPLAIN, GAR-500 and JRC-500) varies with different geographic regions. A Kruskal-Wallis test showed that there is a significant difference in model agreement between the regions, H(6) =146, p < 0.001. The pairwise comparisons, for detecting significant differences between the regions, have been conducted using the Wilcoxon rank-sum test with continuity correction.**

| | n | Mean | S.D. | (1) | (2) | (3) | (4) | (5) | (6) |
|---|---|---|---|---|---|---|---|---|---|
| 1. Africa | 662 | 0.323 | 0.162 | - | | | | | |
| 2. Europe | 327 | 0.365 | 0.162 | *** | - | | | | |
| 3. Siberia | 132 | 0.323 | 0.133 | n.s. | n.s. | - | | | |
| 4. Asia | 515 | 0.387 | 0.198 | *** | n.s. | ** | - | | |
| 5. Australia | 216 | 0.354 | 0.166 | ** | n.s. | n.s. | n.s. | - | |
| 6. South America | 454 | 0.392 | 0.181 | *** | n.s. | *** | n.s. | * | - |
| 7. North America | 470 | 0.291 | 0.149 | *** | *** | *** | *** | *** | *** |

Note. *p < .05, **p < .01, ***p < .001

**Table A4: Summary statistics and pairwise comparisons of how MAI-500 (the model agreement index of GFPLAIN, GAR-500 and JRC-500) varies with different river stream orders. A Kruskal-Wallis test showed that there is a significant difference in model agreement between the river stream orders, H(3) =101, p < 0.001. The pairwise comparisons, for detecting significant differences**
**between the stream orders, have been conducted using the Wilcoxon rank-sum test with continuity correction.**

| | n | Mean | S.D. | (1) | (2) | (3) |
|---|---|---|---|---|---|---|
| 1. Stream order 0 (coast) | 420 | 0.283 | 0.169 | - | | |
| 2. Stream order 1 (inland) | 1147 | 0.372 | 0.184 | *** | - | |
| 3. Stream order 2 (inland) | 910 | 0.355 | 0.167 | *** | n.s. | - |
| 4. Stream order 3 (inland) | 299 | 0.327 | 0.123 | *** | ** | n.s. |

Note. *p < .05, **p < .01, ***p < .001


**Table A5: Summary statistics and pairwise comparisons of how MAI-500 (the model agreement index of GFPLAIN, GAR-500 and JRC-500) varies with the freshwater major habitat types that spatially dominate the river basin. A Kruskal-Wallis test showed that there is a significant difference in model agreement between freshwater habitat types, H(10) =478, p < 0.001.  The pairwise comparisons, for detecting significant differences between the habitat types, have been conducted using the Wilcoxon rank-sum test with continuity correction.**

| | n | Mean | S.D. | (1) | (2) | (3) | (4) | (5) | (6) | (7) | (8) | (9) | (10) |
|---|---|---|---|---|---|---|---|---|---|---|---|---|---|
| 1. Large lakes | 87 | 0.286 | 0.140 | - | | | | | | | | | |
| 2. Large river deltas | 37 | 0.532 | 0.232 | *** | - | | | | | | | | |
| 3. Montane freshwaters | 65 | 0.286 | 0.129 | n.s. | *** | - | | | | | | | |
| 4. Xeric freshwaters and endorheic basins | 304 | 0.260 | 0.176 | * | *** | * | - | | | | | | |
| 5. Temperate coastal rivers | 424 | 0.258 | 0.118 | n.s. | *** | n.s. | n.s. | - | | | | | |
| 6. Temperate upland rivers | 164 | 0.321 | 0.146 | n.s. | *** | n.s. | *** | *** | - | | | | |
| 7. Temperate floodplain rivers and wetlands | 494 | 0.429 | 0.180 | *** | * | *** | *** | *** | *** | - | | | |
| 8. Tropical and subtropical coastal rivers | 366 | 0.343 | 0.149 | *** | *** | ** | *** | *** | n.s. | *** | - | | |
| 9. Tropical and subtropical upland rivers | 303 | 0.388 | 0.149 | *** | *** | *** | *** | *** | *** | ** | *** | - | |
| 10. Tropical and subtropical floodplain rivers and wetlands | 433 | 0.408 | 0.182 | *** | ** | *** | *** | *** | *** | * | *** | n.s. | - |
| 11. Polar freshwaters | 99 | 0.309 | 0.117 | n.s. | *** | n.s. | *** | *** | n.s. | *** | n.s. | *** | *** |

Note. *p < .05, **p < .01, ***p < .001





**Table A6: Total country population (millions) using data from World Bank (The World Bank, 2021a), GHS, HRSL and WorldPop for 26 countries.**

| | World Bank 2015 | GHS 2015 | World Bank 2018 | HRSL 2018 | WorldPop 2018 |
|---|---|---|---|---|---|
| Bangladesh | 156.3 | 175.7 | 161.4 | 170.4 | 137.1 |
| Bolivia | 10.9 | 11.1 | 11.4 | 11.4 | 9.8 |
| Brazil | 204.5 | 219.4 | 209.5 | 219.2 | 181.8 |
| Cambodia | 15.5 | 15.8 | 16.2 | 16.5 | 15.4 |
| Cent. Afr. Rep. | 4.5 | 4.9 | 4.7 | 5.2 | 4.4 |
| Colombia | 47.5 | 48.5 | 49.7 | 51.2 | 51.3 |
| Congo | 4.9 | 4.6 | 5.2 | 4.9 | 3.2 |
| Ecuador | 16.2 | 15.9 | 17.1 | 17.1 | 14.6 |
| Ghana | 27.8 | 27.6 | 29.8 | 29.3 | 26.8 |
| Guatemala | 15.6 | 17.0 | 16.3 | 17.4 | 15.0 |
| Honduras | 9.1 | 8.3 | 9.6 | 8.6 | 7.9 |
| India | 1310.2 | 1408.3 | 1352.6 | 1376.8 | 1158.0 |
| Indonesia | 258.4 | 253.7 | 267.7 | 269.4 | 232.2 |
| Kenya | 47.9 | 46.4 | 51.4 | 49.0 | 44.3 |
| Laos | 6.7 | 6.9 | 7.1 | 6.8 | 6.3 |
| Liberia | 4.5 | 4.5 | 4.8 | 4.8 | 3.7 |
| Malawi | 16.7 | 18.0 | 18.1 | 18.3 | 15.0 |
| Mozambique | 27.0 | 28.8 | 29.5 | 29.7 | 24.8 |
| Nicaragua | 6.2 | 6.2 | 6.5 | 6.5 | 5.7 |
| Nigeria | 181.1 | 184.0 | 195.9 | 193.6 | 174.9 |
| Peru | 30.5 | 32.0 | 32.0 | 33.3 | 29.3 |
| Sri Lanka | 21.0 | 20.8 | 21.7 | 21.9 | 18.1 |
| Thailand | 68.7 | 70.4 | 69.4 | 72.3 | 62.7 |
| Uganda | 38.2 | 39.1 | 42.7 | 41.5 | 34.1 |
| Tanzania | 51.5 | 53.4 | 56.3 | 56.7 | 45.7 |
| Viet Nam | 92.7 | 97.4 | 95.5 | 99.1 | 82.6 |




**Table A7: Total land area and flood-prone area (square kilometres) using GFPLAIN, JRC-100, JRC-500, GAR-100 and GAR-500. Land area is the total country area minus surface water area, given by the water mask MOD44W. The percentage of country area that is normally occurring surface water is less than 5 % for all countries, except Uganda with Lake Victoria (15 %) and Malawi with Lake Malawi (20 %).**

|  | Total land | GFPLAIN | GAR-100 | GAR-500 | JRC-100 | JRC-500 |
|---|---|---|---|---|---|---|
| Bangladesh | 133255 | 75291 | 81469 | 85451 | 58237 | 61949 |
| Bolivia | 1073122 | 283994 | 226717 | 234136 | 108318 | 119279 |
| Brazil | 8403158 | 963993 | 1133095 | 1161675 | 549877 | 594106 |
| Cambodia | 177699 | 49628 | 48407 | 50342 | 27490 | 29069 |
| Cent. Afr. Rep. | 621613 | 49434 | 41201 | 41738 | 20128 | 22344 |
| Colombia | 1130606 | 172221 | 178504 | 181772 | 87246 | 94759 |
| Congo | 339743 | 66827 | 45277 | 46018 | 23487 | 27041 |
| Ecuador | 255003 | 14367 | 25775 | 26306 | 5841 | 6444 |
| Ghana | 233261 | 25815 | 18243 | 18485 | 6603 | 7288 |
| Guatemala | 108183 | 14325 | 14090 | 14584 | 3590 | 3898 |
| Honduras | 111452 | 7458 | 15194 | 15780 | 7255 | 7947 |
| India | 2940760 | 385664 | 354450 | 375427 | 251512 | 274302 |
| Indonesia | 1862962 | 253940 | 268564 | 278633 | 110383 | 121199 |
| Kenya | 572913 | 72052 | 52158 | 54522 | 21850 | 24488 |
| Laos | 229519 | 23940 | 32232 | 33597 | 10586 | 11353 |
| Liberia | 96230 | 8252 | 12316 | 12471 | 1812 | 1908 |
| Malawi | 95161 | 7325 | 7611 | 7721 | 2034 | 2238 |
| Mozambique | 775817 | 70647 | 76199 | 78920 | 32177 | 34706 |
| Nicaragua | 118971 | 12616 | 21336 | 22389 | 5716 | 6312 |
| Nigeria | 905864 | 170522 | 110735 | 115619 | 73731 | 80413 |
| Peru | 1281244 | 136202 | 152788 | 155400 | 79743 | 88004 |
| Sri Lanka | 64722 | 3898 | 6811 | 7086 | 693 | 770 |
| Tanzania | 883844 | 112606 | 75055 | 77959 | 22289 | 24367 |
| Thailand | 511152 | 105223 | 97805 | 106312 | 57068 | 59844 |
| Uganda | 205777 | 25341 | 16703 | 17428 | 5700 | 6231 |
| Viet Nam | 323857 | 62585 | 72180 | 75512 | 53951 | 55392 |



**Table A8: Population estimates within flood-prone areas (millions) intersecting GFPLAIN, GAR-500 and JRC-500 with GHS, HRSL and WorldPop for 26 countries.**

| | GHS | | | HRSL | | | WorldPop | | |
|---|---|---|---|---|---|---|---|---|---|
| | GFPLAIN | GAR-500 | JRC-500 | GFPLAIN | GAR-500 | JRC-500 | GFPLAIN | GAR-500 | JRC-500 |
| Bangladesh | 104.1 | 126.0 | 84.5 | 100.5 | 121.4 | 83.5 | 81.9 | 98.5 | 70.0 |
| Bolivia | 2.1 | 2.1 | 0.6 | 1.8 | 1.6 | 0.4 | 1.5 | 1.4 | 0.4 |
| Brazil | 28.8 | 36.3 | 11.3 | 29.1 | 36.6 | 11.7 | 24.6 | 30.6 | 9.9 |
| Cambodia | 8.4 | 8.8 | 5.5 | 8.7 | 9.2 | 5.7 | 7.9 | 8.0 | 5.4 |
| Cent. Afr. Rep. | 1.3 | 0.9 | 0.4 | 1.4 | 1.0 | 0.4 | 1.1 | 0.8 | 0.4 |
| Colombia | 14.4 | 10.5 | 3.4 | 13.8 | 9.3 | 3.0 | 13.0 | 9.3 | 3.1 |
| Congo | 1.1 | 1.4 | 0.8 | 1.1 | 1.5 | 0.7 | 1.0 | 0.7 | 0.6 |
| Ecuador | 3.3 | 4.5 | 1.3 | 3.1 | 4.3 | 1.3 | 2.5 | 3.5 | 1.1 |
| Ghana | 2.3 | 2.3 | 0.9 | 1.7 | 1.9 | 0.4 | 1.8 | 1.7 | 0.5 |
| Guatemala | 1.6 | 2.5 | 0.6 | 0.9 | 1.4 | 0.2 | 0.8 | 1.2 | 0.2 |
| Honduras | 0.8 | 1.6 | 0.5 | 0.6 | 1.2 | 0.4 | 0.6 | 1.2 | 0.4 |
| India | 379.9 | 337.7 | 251.6 | 345.6 | 301.5 | 231.2 | 306.9 | 260.8 | 213.6 |
| Indonesia | 43.7 | 45.0 | 20.7 | 37.2 | 40.7 | 19.1 | 31.1 | 34.0 | 15.9 |
| Kenya | 3.8 | 3.8 | 1.1 | 3.3 | 3.8 | 0.7 | 3.1 | 3.3 | 0.8 |
| Laos | 4.0 | 3.5 | 2.4 | 3.4 | 3.2 | 2.0 | 2.7 | 2.7 | 1.6 |
| Liberia | 0.5 | 1.1 | 0.9 | 0.5 | 1.2 | 0.8 | 0.4 | 0.9 | 0.8 |
| Malawi | 2.0 | 2.0 | 0.5 | 1.6 | 1.9 | 0.4 | 1.4 | 1.5 | 0.4 |
| Mozambique | 4.9 | 5.4 | 2.6 | 2.9 | 3.3 | 1.3 | 2.7 | 2.8 | 1.7 |
| Nicaragua | 0.8 | 1.1 | 0.3 | 0.4 | 0.8 | 0.1 | 0.4 | 0.7 | 0.1 |
| Nigeria | 37.7 | 29.1 | 21.2 | 31.7 | 23.1 | 17.3 | 31.0 | 23.4 | 18.2 |
| Peru | 4.2 | 6.2 | 2.0 | 3.6 | 5.5 | 1.7 | 2.8 | 4.7 | 1.3 |
| Sri Lanka | 2.7 | 3.9 | 0.1 | 2.6 | 4.1 | 0.1 | 2.3 | 3.5 | 0.1 |
| Thailand | 34.7 | 37.4 | 27.7 | 33.3 | 36.1 | 27.3 | 29.5 | 31.7 | 25.2 |
| Uganda | 3.2 | 3.0 | 0.6 | 2.8 | 2.6 | 0.3 | 3.2 | 2.5 | 0.6 |
| Tanzania | 6.8 | 4.6 | 1.4 | 5.4 | 3.5 | 0.5 | 4.7 | 3.1 | 0.7 |
| Viet Nam | 54.9 | 58.4 | 42.7 | 53.4 | 57.4 | 43.6 | 43.5 | 46.0 | 35.8 |



**Table A9: Population estimates within flood-prone areas (millions) using a return period of 100 years compared to 500 years, intersecting GAR and JRC with the HRSL population dataset.**

|  | GAR-100 | GAR-500 | JRC-100 | JRC-500 |
|---|---|---|---|---|
| Bangladesh | 116.14 | 121.38 | 76.99 | 83.52 |
| Bolivia | 1.48 | 1.56 | 0.36 | 0.39 |
| Brazil | 31.37 | 36.57 | 10.77 | 11.71 |
| Cambodia | 8.93 | 9.21 | 5.38 | 5.69 |
| Cent. Afr. Rep. | 0.94 | 0.96 | 0.38 | 0.44 |
| Colombia | 8.25 | 9.29 | 2.71 | 2.95 |
| Congo | 1.46 | 1.53 | 0.64 | 0.70 |
| Ecuador | 3.93 | 4.28 | 1.20 | 1.32 |
| Ghana | 1.83 | 1.91 | 0.38 | 0.41 |
| Guatemala | 1.34 | 1.37 | 0.21 | 0.23 |
| Honduras | 1.17 | 1.22 | 0.35 | 0.37 |
| India | 279.11 | 301.54 | 208.74 | 231.19 |
| Indonesia | 33.39 | 40.71 | 18.05 | 19.11 |
| Kenya | 3.67 | 3.81 | 0.63 | 0.75 |
| Laos | 3.05 | 3.23 | 1.87 | 1.96 |
| Liberia | 1.14 | 1.16 | 0.76 | 0.81 |
| Malawi | 1.82 | 1.85 | 0.31 | 0.38 |
| Mozambique | 3.18 | 3.26 | 1.17 | 1.29 |
| Nicaragua | 0.72 | 0.76 | 0.12 | 0.13 |
| Nigeria | 22.21 | 23.14 | 14.97 | 17.33 |
| Peru | 4.94 | 5.55 | 1.63 | 1.74 |
| Sri Lanka | 3.85 | 4.13 | 0.04 | 0.05 |
| Thailand | 19.75 | 36.08 | 26.43 | 27.34 |
| Uganda | 2.28 | 2.58 | 0.24 | 0.29 |
| Tanzania | 3.36 | 3.48 | 0.43 | 0.49 |
| Viet Nam | 50.76 | 57.42 | 41.69 | 43.58 |




**Code availability**

The geospatial analysis has primarily been conducted in Google Earth Engine, and the statistical analysis has been conducted
in R. The corresponding codes are made available upon request.

**Data availability**

The floodplain layer GFPLAIN250m is available at https://figshare.com/articles/GFPLAIN250m/6665165/1. The JRC Flood
hazard map of the World is available at https://data.jrc.ec.europa.eu/collection/id-0054. The Global Assessment Report 2015
flood hazard maps can be accessed via the PREVIEW Global Risk Data Platform https://preview.grid.unep.ch/. HydroBASINS
and HydroATLAS are available at https://www.hydrosheds.org/. The remaining datasets were all accessed through the Google
Earth Engine Data Catalog (Gorelick et al., 2017):

- Global Administrative Unit Layers: ee.FeatureCollection("FAO/GAUL/2015/level0")
- Global Human Settlement Layer: ee.ImageCollection("JRC/GHSL/P2016/POP_GPW_GLOBE_V1")
- High Resolution Settlement Layer: ee.ImageCollection("projects/sat-io/open-datasets/hrslpop")
- HydroBASINS: ee.FeatureCollection("WWF/HydroSHEDS/v1/Basins/hybas_5")
- MOD44W Land Water Mask: ee.Image("MODIS/MOD44W/MOD44W_005_2000_02_24")
- WorldPop: ee.ImageCollection("WorldPop/GP/100m/pop")

**Author contribution**

SL, JM, LB and GDB contributed to the study conceptualizations. SL designed and carried out the data analysis and
visualization. SL prepared the original manuscript draft. JM, LB and GDB reviewed and edited the manuscript.

**Competing interests**

The authors declare that they have no conflict of interest.

**Acknowledgements**

This work was financially supported by the European Commission, Programme: H2020 Excellent Science - European Research
Council, Grant/Award Number: 771678. The authors acknowledge Prof. Fernando Nardi for providing the floodplain layer
GFPLAIN250m, and the European Commission's JRC and CIMA/UNEP for providing the global flood hazard maps. We also
acknowledge the European Commission, Facebook Connectivity Lab with Columbia University and WorldPop for providing
the population datasets.



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
