# Peer review of "Global riverine flood risk – how do hydrogeomorphic floodplain maps compare to flood hazard maps?"

_Natural Hazards and Earth System Sciences, 2021_

## Author Response (AR1)

**Response to referee RC1**

The author team is grateful to referee RC1 for writing a very helpful, concise and constructive review that addresses multiple points. We believe that, by carefully addressing all of them, we substantially improved our research article. Please find answers to the individual points of improvement below:

- **RC1**: *"In the abstract it is stated that flood mapping can be based on either (hydrologically-derived) flood hazard maps or (topography-based) hydrogeomorphic floodplain maps: what about satellite-derived?"*
  **Authors**: Thanks for noting this. Indeed, satellite imagery is gaining importance for flood mapping. Yet, our focus is on these two methods since satellite imagery can miss some areas that were not flooded in the observation period (or covered by clouds), but can still be *potentially* flooded in the future. We added the word "modelling" (L12) to clarify that we mean mapping areas that are prone to potential flooding, rather than mapping areas that have flooded in the past.

- **RC1**: *"In the introduction (L25-28) it is stated that global maps of flood-prone zones and human settlements are useful for detecting risk hotspots across the world, and may also be used for local studies in data-poor regions. The following papers are cited: UN SDSN, 2020; Ward et al., 2020. In fact the paper of Ward et al. 2020 does not seem so relevant here, but the community perspective paper on this issue from 2015 may be (https://www.nature.com/articles/nclimate2742)"*
  **Authors**: We agree with RC1 on this point and have changed this accordingly (L28).

- **RC1**: *"L38-39: "Hydrogeomorphic methods for mapping floodplains, on the other hand, distinguish the characteristic shapes of floodplains based on topography". This feels like a non sequitur: in the previous sentence a lack of detailed topographic data was mentioned as a main limitations to model water extent – but here it appears that these same data are needed for the hydrogeomorphic method. Or is there a difference in the scale/type of data needed between the two methods?"*
  **Authors**: We agree with RC1 that these two sections need to be rephrased, we have changed the text (L35-40).

- **RC1**: *"L50-51: "What we know, however, is that the results of large-scale flood exposure analysis heavily depend upon the datasets used (Aerts et al., 2020; Dottori et al., 2016; Smith et al., 2019; Trigg et al., 2016; Ward et al., 2020)." The study of Bernhofen et al. should be included in this list."*
  **Authors**: We agree with RC1 that Bernhofen et al should be included here, we have added this reference here (L52).

- **RC1**: *"L68: mention the names of the 2 flood hazards datasets here (before the references to Dottori and CIMA)"*
  **Authors**: Thanks for noting this, we have changed this in the text (L69-70).

- **RC1**: *"I do not understand the motivation for using 3 different population maps if the focus is on comparing hazard modelling, as implied by the title. For me the message can be obtained by selecting one of these datasets and the analysis would be more focused."*
  **Authors**: Thanks for raising this important point, indeed, including only one population dataset would possibly make the analysis more focused. However, we think that including the three population datasets makes the exposure analysis more comprehensive and transparent. The individual population datasets have different limitations and benefits, and all three are frequently used within the academic community working on disaster analysis. We, therefore, decided that it would be relevant to include all three datasets, to show differences in exposure results. We agree with RC1 that this needs to be justified more in the text, we have added some text in section 2.2 (L132-134).

- **RC1**: *"Section 3.1: Am I correct to assume that all maps were homogenized to 8.33 arcsecs? I could not find this explicitly in the text. Is this correct? And if so make this clear in section 3.1"*
  **Authors**: Yes, this is correct. Thanks for pointing this out, it is now written in section 3.1 (L160-161).

- **RC1**: *"L161-163: "For instance, the individual flood maps have been post-processed to mask arid areas, to different degrees, since aridity poses a challenge for traditional flood model assumptions." I find this very vague. To "What degrees" exactly? How was this done and based on what assumptions? This is essential for reproducibility."*
  **Authors**: Thanks for noting this. This statement referred to the original datasets, we meant that the creators of the flood hazard maps and the floodplain map have post-processed the maps to mask arid areas to different

degrees (i.e. the original datasets vary in spatial coverage). For this reason, we calculated the model agreement index only for the river basins that are covered by all three models. We have rephrased this sentence to clarify (L173).

- **RC1**: *"What was the reason for choosing the 26 countries shown in the analysis?"*
  **Authors**: We based the country selection on four criteria, as specified in section 3.2 (lines 238-252). But indeed, the number of included countries could have been higher or lower. We decided on 26 countries since we thought that this was a large enough sample to provide variation across countries while still presenting the results clearly. We have now also add a fifth selection criterion: that we aimed to get a representative country sample across the regions of the world.

**Response to referee RC2**

The author team is grateful to referee Francesco Dottori for providing a thorough, constructive and helpful review, including several comments both in terms of general and specific issues. We believe that, by carefully addressing all of them, we substantially improved our research article. We have commented on each point below:

**General comments**

- **RC2**: *"In the conclusions, the Authors state that "Inter-model comparisons like this study do not answer the question of how well the individual flood layers agree with actual flood events". I believe that the paper should provide some provide a better description of what we presently know about the skill of global flood maps. So far, global flood models have been validated and compared only in few regions (Bernhofer et al 2018: Sampson et al., 2015) and model skill has shown to be unsatisfactory in some areas due to limitations of global models and data (e.g. Dottori et al. 2016). On top of that, the present study confirmed the limited agreement of global flood maps, meaning that the overall uncertainty is still quite large. I invite the Authors to elaborate further on this topic, in order to put the intercomparison study in perspective."*
  **Authors**: This is a great idea and we agree that elaborating more on this topic would enrich the manuscript. We have added information about previous findings by Bernhofen et al. (2018) how JRC and GAR performed in their validation study (L105-109). We have also added a suggestion for future validation work based on recently released by Hawker et al. (2020) and Tellman et al. (2021) in Section 5 (L580-581). We also wrote how our result that the models show low agreement in deltas are in line with previous findings by Dottori et al. (2016) in Section 4.1.2 (L348).

- **RC2**: *"In light of these considerations, maybe the Authors could also provide suggestions on how the outcomes of their study can be useful for real-world applications (for instance, should we use ensemble of flood maps as usually done for climate projections?)"*
  **Authors**: Thanks for providing this good suggestion. We agree and a suggestion to users in Section 5 (L569-570).

- **RC2**: *"the authors mention that GFPLAIN appoach can be swiftly applied to update floodplain maps whenever new elevation data are available. Perhaps, GFPLAIN could also be applied to map the minor river network (i.e. river basins with area <1000km2), thus providing information on potential flood-prone areas that are generally not included in global flood maps. Perhaps the Autors could elaborate on this point. Do you think that GFPLAIN has the potential to do that, or perhaps there are limitations that could hinder such application?"*
  **Authors**: This is an interesting point. Indeed, with higher spatial resolution of the terrain data comes the opportunity to read finer patterns in the landscape at smaller parts of the river network. For instance, Nardi et al. (2006) used an upper threshold of 7 $km^2$ when working with a 30 m terrain model, Manfreda et al. (2014) used an upper threshold of 0.1 $km^2$ when working with an 80 m terrain model and Tavares da Costa et al. (2019) used an upper threshold of 0.1 $km^2$ when working with a 25 m terrain model. But at the same time, as phrased by Nardi et al. (2018): "floodplain landscapes are primarily created by large-scale hydrologic forces over extended periods of time, as opposed to localized hydraulic interactions", which we suppose would mean a limitation in the ability of the method to capture the very finest parts of the river network. We have briefly highlighted this application opportunity in Section 1 (L41) and 5 (L588).

**Specific comments**

- **RC2**: *"Section 2.1: according to Table A1, GAR flood maps include (at least partially) the effect of dams and reservoirs, this is something I would mention here."*
  **Authors**: Indeed, this is an important point. This difference is described on L110-L116, but we have rewritten this for clarity (L113-114).

- **RC2**: *"L159-160: "MAI values, Eq. (1), were then calculated for all the basins in the world that are covered by all three models, resulting in 2776 river basins". My understanding is that larger river basins (e.g. Amazon, Mississippi etc) have been split into sub-basins for this analysis, correct? Can you please provide some information about the average-min-max areas of the river basins analyzed?"*
  **Authors**: Yes, this understanding is correct. We have provided this information briefly in the text (L175-177), and in a new Table A2.

- **RC2**: *"L230: typo (Vietnam)"*
  **Authors**: We used the spelling "Viet Nam" throughout the manuscript and figures, as it is still used by the United Nations (https://unsdg.un.org/un-in-action/viet-nam) and by the Vietnamese Government.

- **RC2**: *"Figure 1 caption: perhaps "Flood exposure in the 26 countries...." is more appropriate here"*
  **Authors**: We agree and have changed this.

- **RC2**: *"L305-307: my understanding here is that spatial distribution of MAI-500 is calculated irrespective of basin area (e.g. small basins count as large basins). Could you please add some justification for this approach?"*
  **Authors**: Yes, you are right. The spatial distribution is based on a local spatial autocorrelation analysis identifying clusters of basins having higher-than-average or lower-than-average model agreement (as described in section 3.2.1. We have added justification for this (L209-210).

- **RC2**: *"L305-307: "Figure 3 provides the spatial distribution of MAI-500 across all 2776 river basins, and local clusters of high and low model agreement basins as identified from the spatial autocorrelation analysis". Could you please specify how "high" and "low" model agreement are defined here?"*
  **Authors**: This is described in section 3.2.1 (L203-207), but we have briefly repeated this in section 4.1.2 (L323) for clarity.

- **RC2**: *"Figures 2 and 4: please describe in the caption the meaning of all graphical elements (e.g. do boxes represent standard deviation or quantiles? meaning of crosses and yellow lines, etc)"*
  **Authors**: Thanks for noting this, we have added this to both figure captions.

- **RC2**: *"Figure 3: please add in the legend the meaning of gray areas"*
  **Authors**: Thanks for noting this, we have added this to the legend of Figure 3.

- **RC2**: *"Figure 3 caption: I would delete lines 314-315, these are comments to results that are already in the main text"*
  **Authors**: We agree and have changed the caption accordingly.

- **RC2**: *"L320: "The snow and ice regions in North America" is not a great definition, perhaps "The regions in North America where river flow is influenced by snow accumulation and snow melt", or just "The regions in North America with cold climate"."*
  **Authors**: Thanks for pointing this out, we have changed this.

- **RC2**: *"L324; "This can, at least partly, be explained by the same regions being dry in the sense that they are snow-covered". Cold or mountain regions can have a dry climate irrespective of snow cover (it is indeed the case of western sector of southern Andes), please rephrase."*
  **Authors**: Thanks for pointing this out, you are right. We have changed it (L337-338)

- **RC2**: *"L330-332: "A possible explanation for the low agreement in coastal river basins might be that the individual riverine flood maps differ in how they mask coastal areas. For instance, GFPLAIN tends to mask areas near the coast, while JRC does not." This is not much clear to me. Could you please briefly explain why and how coastal areas are masked out (or not detected) in GFPLAIN flood maps? JRC maps also could miss or understimate smaller coastal basins due to their high threshold on upstream area."*
  **Authors**: Thanks for noting this, we agree. We have clarified these points in the text (L347-355).

- **RC2**: *"Figure 4a: Why is Siberia included here while Oceania is not? Is it an oversight or done on purpose?"*
  **Authors**: Thanks for pointing this out, this is an oversight. We used the geographical regions as categorised

by the HydroBasins dataset, which originally had the name "Australia and Oceania". We have changed this in Figure 4 and Table A4.

- **RC2**: *"Figure 4b: Which definition of stream order are you using? How are coastal basins defined exactly? I looked at the related references but could not find an explanation on these points, so please provide some details"*
  **Authors**: Good point. Information about this can be found in the Technical Documentation of HydroBasins., we have added that as a reference and briefly described the ordering system in Table 2.

- **RC2**: *"Figure 5: I don't see the reason for including pair-wise correlation between all variables, I suggest leaving only MAI against the other variables (this would also improve the readability of correlation values between -0.25 and 0.25)"*
  **Authors**: This is a good suggestion, and we agree that this would improve the focus and readability of the figure. We have changed Figure 5 accordingly, and also wrote out the values to improve the readability.

- **RC2**: *"Section 4.1.3: my impression is that the the discussion of correlation values is not fully consistent with what shown in Figure 5 (perhaps because the color scale makes values not easy to distinguish, see my comment above). For instance, it seems that correlations between model agreement and some anthropogenic influence factors (e.g. population count) are stronger than for some climatic factors (e.g. annual precipitation)"*
  **Authors**: Thanks for raising this point, we have changed Figure 5 to improve the readability (see answer above) and have also revised the discussion in section 4.1.3 to ensure consistency (L444-446).

- **RC2**: *"figure 7: please clarify in the legend (or in the caption) how exposed population bars should be read (e.g. GFPLAIN values on the left, GAR value as shaded bars on the right etc)"*
  **Authors**: We agree that this needs to be clarified, we have clarified this in the figure caption.

- **RC2**: *"caption figure 8: I would delete the part: " GFPLAIN and GAR cover ... for the Niger Delta (b)", these are comments to results that fit better in the main text"*
  **Authors**: Yes, you are right. We have deleted it in the caption.